# Morphological Changes of 3T3 Cells under Simulated Microgravity

**DOI:** 10.3390/cells13040344

**Published:** 2024-02-15

**Authors:** Minh Thi Tran, Chi Nguyen Quynh Ho, Son Nghia Hoang, Chung Chinh Doan, Minh Thai Nguyen, Huy Duc Van, Cang Ngoc Ly, Cuong Phan Minh Le, Huy Nghia Quang Hoang, Han Thai Minh Nguyen, Han Thi Truong, Quan Minh To, Tram Thi Thuy Nguyen, Long Thanh Le

**Affiliations:** 1Faculty of Applied Technology, School of Technology, Van Lang University, Ho Chi Minh City 70000, Vietnam; minh.tt@vlu.edu.vn; 2Animal Biotechnology Department, Institute of Tropical Biology, Vietnam Academy of Science and Technology, Ho Chi Minh City 70000, Vietnam; quynhchihonguyen@gmail.com (C.N.Q.H.); hoangsonitb@gmail.com (S.N.H.); doanchinhchung@gmail.com (C.C.D.); tminh0602@gmail.com (M.T.N.); huy.biotech@gmail.com (H.D.V.); lyngoccang@gmail.com (C.N.L.); lephanminhcuong@gmail.com (C.P.M.L.); hoangnghiaquanghuy@gmail.com (H.N.Q.H.); han.nguyen@unh.edu (H.T.M.N.); ttramnt@gmail.com (T.T.T.N.); 3Biotechnology Department, Graduate University of Science and Technology, Vietnam Academy of Science and Technology, Ha Noi City 100000, Vietnam; 4Biotechnology Innovation Center, University of New Hampshire, Manchester, NH 03101, USA; 5Department of Biophysics, Sungkyunkwan University, Suwon 16419, Republic of Korea; hantruong029@g.skku.edu; 6Faculty of Biology and Biotechnology, University of Science, Ho Chi Minh City 70000, Vietnam; tomquan@hcmus.edu.vn; 7Faculty of General Biomedical, University of Physical Education and Sport, Ho Chi Minh City 70000, Vietnam

**Keywords:** 3T3 cell, cell cycle progression, cytokinesis, cytoskeleton, morphology, simulated microgravity

## Abstract

Background: Cells are sensitive to changes in gravity, especially the cytoskeletal structures that determine cell morphology. The aim of this study was to assess the effects of simulated microgravity (SMG) on 3T3 cell morphology, as demonstrated by a characterization of the morphology of cells and nuclei, alterations of microfilaments and microtubules, and changes in cycle progression. Methods: 3T3 cells underwent induced SMG for 72 h with Gravite^®^, while the control group was under 1G. Fluorescent staining was applied to estimate the morphology of cells and nuclei and the cytoskeleton distribution of 3T3 cells. Cell cycle progression was assessed by using the cell cycle app of the Cytell microscope, and Western blot was conducted to determine the expression of the major structural proteins and main cell cycle regulators. Results: The results show that SMG led to decreased nuclear intensity, nuclear area, and nuclear shape and increased cell diameter in 3T3 cells. The 3T3 cells in the SMG group appeared to have a flat form and diminished microvillus formation, while cells in the control group displayed an apical shape and abundant microvilli. The 3T3 cells under SMG exhibited microtubule distribution surrounding the nucleus, compared to the perinuclear accumulation in control cells. Irregular forms of the contractile ring and polar spindle were observed in 3T3 cells under SMG. The changes in cytoskeleton structure were caused by alterations in the expression of major cytoskeletal proteins, including β-actin and α-tubulin 3. Moreover, SMG induced 3T3 cells into the arrest phase by reducing main cell cycle related genes, which also affected the formation of cytoskeleton structures such as microfilaments and microtubules. Conclusions: These results reveal that SMG generated morphological changes in 3T3 cells by remodeling the cytoskeleton structure and downregulating major structural proteins and cell cycle regulators.

## 1. Introduction

Gravity plays an important role in the growth and development of organisms [1]. Gravity also affects the formation of morphological characteristics of living organisms on Earth, from tissue structure to cells [2]. An early study showed that an asymmetric distribution of microtubules was seen in the early development of shellfish under induced microgravity, which resulted in abnormal morphological development [3]. Another study reported that gravity generated lymphocyte deformation through changed cell shape [4]. The shape of bone cells changed dramatically after exposure to microgravity compared to control cells [5]. Morphological and phenotypic transitions were shown in MCF7 cells under simulated microgravity [6]; the changes in cell morphology were generated by alterations in the cytoskeletal structure. The actin cytoskeleton of osteoblasts exposed to microgravity for 4 days entirely collapsed [7]. Real microgravity induced disorganization in breast cancer MCF-7 cells and the thyroid cancer FTC-133 cell line [8,9]. However, another study showed that there was no change in actin structure and organization upon exposure to SMG [10]. Other investigations have found enhanced formation of actin and stress fibers associated with the development of lamellipodia protrusions under microgravity conditions [11,12]. Beta-tubulin levels of activated T cells increased markedly during microgravity treatment [13]. 

Therefore, although changes in cell morphology and cytoskeletal structure have been well characterized, there is still much controversy regarding the effects of microgravity on the formation and distribution of cell structures. This may be due to experiments using different conditions, such as real or simulated microgravity, and different microgravity simulators, such as 2D clinostat, 3D clinostat, or rotating vessel. The morphological changes exhibit close associations with cell proliferation and cell cycle progression [14,15]. However, the effects of SMG on these processes, such as morphological alterations and cell cycle related protein expression, have not been well characterized. Therefore, in this study, we used the Gravite^®^ 3D clinostat to simulate microgravity and assess the effects of SMG on cell morphology, as demonstrated by modifications in cell nucleus and cytoplasm, especially cytoskeleton distribution, changes in cell cycle progression and cell division structures, and the expression of major cytoskeletal proteins.

## 2. Materials and Methods

### 2.1. Cell Culture and SMG Induction

In this study, 3T3 cells (a gift of Dr. Thuan Lao, Open University, Ho Chi Minh City) were cultured in DMEM/Ham’s F-12 (DMEM-12-A) with 15% FBS (FBS-HI-22B) and 1% Pen/Strep (PS-B) (all from Capricorn Scientific, Ebsdorfergrund, Germany). The 3T3 cells were cultured in T-25 flasks and 96-well plates (160430 and 161093; Thermo Fisher Scientific, Inc., Waltham, MA, USA). The culture medium was poured into the flasks and wells without bubbles to avoid the shearing of fluid [16]. The flasks and plates were fixed in the stage of Gravite^®^ gravity controller (AS ONE International, Inc., Santa Clara, CA, USA). The Gravite^®^ was placed in a CO_2_ incubator (MCO-18AIC, Sanyo Electric Co., Osaka, Japan) (Figure 1E). The Gravite offers 4 microgravity simulation programs: mode A: ×4 rpm; mode B: ×3 rpm; mode C: ×2 rpm; and mode D: ×1 rpm. Of these, mode C is advised for cell culture. Thus, 3T3 cells were subjected to simulated microgravity (mode C of Gravite^®^) for 72 h. The control group was under 1G in the same CO_2_ incubator. 

### 2.2. Cell Density Analysis

For this analysis, 3T3 cells were seeded in 96-well plates at a density of 3 × 10^3^ cells/well. The culture medium was added to 400 µL for each well. Wells were covered with parafilm tape. The 3T3 cells underwent SMG for 72 h. The control group was under 1G in the same CO_2_ incubator. After 72 h, the culture medium was discarded. The nuclei of 3T3 cells were stained with Hoechst 33342 (14533, Sigma-Aldrich, St. Louis, MO, USA) for 30 min. The 3T3 cells were washed 3 times with phosphate-buffered saline (Gibco, Thermo Fisher Scientific, Inc., Waltham, MA, USA). Cells were observed under fluorescence with a Cytell cell imaging system (GE Healthcare, Chicago, IL, USA) and cell number was determined by counting nuclei with the Cytell’s cell cycle app. The cell cycle app was also used to evaluate cell cycle progression and nuclear morphology parameters, including nuclear intensity, shape, and area. In order to assess the nuclear properties of the 3T3 cells, the parameter of nuclear area (µm^2^) was adjusted to 150 and the parameter of sensitivity (%) was adjusted to 50 (according to the Cytell manual). 

### 2.3. Flow Cytometry Analysis

For this analysis, 3T3 cells were seeded in T-25 flasks at a density of 1 × 10^5^ cells/flask, then underwent SMG for 72 h. The control group was under 1G in the same CO_2_ incubator. After 72 h, the culture medium was discarded, and cell detachment was performed with 1 mL trypsin-EDTA 0.25% (TRY-2B, Capricorn Scientific, Ebsdorfergrund, Germany). Cells were collected to 1.5 mL of PBS and the FSC value was subsequently analyzed using a BD Accuri C6 Plus flow cytometer (BD Biosciences, San Jose, CA, USA).

### 2.4. Western Blot

For this analysis, 3T3 cells were plated in T-25 flasks at a density of 1 × 10^5^ cells/flask and underwent induced SMG for 72 h. The control group was under 1G in the same CO_2_ incubator. The 3T3 cells were collected from the flasks, and the lysate was treated with LDS Sample Buffer (ab119196;). The protein samples were loaded to SDS-PAGE gel (ab139596;) in equal amounts. Gel running was performed with running buffer (ab119197) at 50 V for 2 h. Protein samples were transferred to PVDF membrane (ab133411) at 90 V for 2 h. The membrane was treated with blocking buffer (ab126587) for 1 h at room temperature. The membrane was incubated with primary antibody overnight at 4 °C. Anti-β actin antibody (ab8226) and anti-α tubulin 3 antibody (ab52866) were used at 1:10,000 dilution, and anti-Cdk4 antibody (ab137675) and anti-Cdk6 antibody (ab124821) were used at 1:5000 dilution. Anti-GAPDH antibody (ab181602) was used as the control at 1:10,000 dilution. The membrane was washed 3 times with TBST for 10 min each time. The membrane was incubated with secondary antibody at room temperature for 1 h. Goat anti-mouse IgG (HRP) (ab6789) was used against beta actin antibody and goat anti-rabbit IgG (HRP) (ab6721) was used against other primary antibodies. An ECL Kit (ab65623) was used to visualize the blots, and imaging was carried out with X-ray film. (All products and materials used for Western blot were from Abcam, Cambridge, MA, USA). 

### 2.5. Microtubule Staining

For this analysis, 3T3 cells were seeded in 96-well plates at a density of 1 × 10^3^ cells/well. The 3T3 cells were treated with 50 nM SiR-tubulin (CY-SC002, Cytoskeleton, Inc., Denver, CO, USA) in each well and underwent SMG for 72 h. The control group was under 1G in the same CO_2_ incubator. After 72 h, the culture medium was discarded. The structure and distribution of microtubules was evaluated under the Cytell microscope (GE Healthcare, Chicago, IL, USA). 

### 2.6. Microfilament Staining 

For this analysis, 3T3 cells were seeded in 96-well plates at a density of 1 × 10^3^ cells/well and underwent SMG for 72 h. The control group was under 1G in the same CO_2_ incubator. After 72 h, the 3T3 cells were fixed with 4% paraformaldehyde (Nacalai Tesque, Kyoto, Japan) for 30 min, then permeabilized with 0.1% Triton X-100 (Merck, Darmstadt, Germany) overnight at 4 °C. Cells were incubated with Phalloidin CruzFluor™ 488 Conjugate (sc-363791; Santa Cruz Biotechnology, Santa Cruz, CA, USA) to stain microfilaments for 1 h. The nuclei were stained with Hoechst 33342 (14533; Sigma-Aldrich, St. Louis, MO, USA) for 30 min. The 3T3 cells were washed 3 times with phosphate-buffered saline (Gibco, Thermo Fisher Scientific, Inc., Waltham, MA, USA) for 10 min each. The distribution of microfilament bundles in 3T3 cells was evaluated under the Cytell microscope. 

### 2.7. Statistical Analysis

The study included three or more replications of each experiment. All data are expressed as the mean ± standard deviation. The data were analyzed for statistical significance by one-way ANOVA, where *p* < 0.05 was considered statistically significant.

## 3. Results

### 3.1. Effects of SMG on 3T3 Cell Size

The present study was conducted to evaluate the effects of SMG on the cell morphology of 3T3 cells. As seen in Figure 1A,B, 3T3 cells in both groups exhibited their normal morphology. Cytoplasmic fragmentation, condensation, and shrinkage, which are characteristics of apoptosis, were not noted in cells from both groups. The 3T3 cells under SMG showed stronger expansion of cytoplasm compared to the 3T3 cells in the control group. To clarify the changes in cell size, flow cytometry was used to determine the FCS value of cells (Appendix A). The results show that the FSC value of 3T3 cells in the control group (9.47 ± 0.44 × 10^6^) was lower compared to the SMG group (10.40 ± 0.15 × 10^6^) (*p* = 0.002) (Figure 1C,D). These results suggest that SMG induced an increase in 3T3 cell size.

### 3.2. Effects of SMG on 3T3 Nuclear Morphology

To estimate nuclear condensation, nuclear intensity was measured by the cell cycle app. The result shows that the nuclear intensity of 3T3 cells in the control group (1.36 ± 0.06/cell) was higher compared to the SMG group (1.30 ± 0.05/cell) (Figure 2A, Appendix A). The nuclear shape value (1.0 = circle, <1.0 = non-circular) was also generated by the cell cycle app. The nuclear shape of 3T3 cells under SMG (0.927 ± 0.003) was lower than that of the control group (0.930 ± 0.003) (Figure 2B, Appendix A), as illustrated by the lower distribution of nuclear shape values of 3T3 cells in the SMG group than the control group (Figure 2D,E). We also found that SMG induced an increase in nuclear size. The nuclear area of 3T3 cells in the SMG group was larger than the that of the control group (215.79 ± 5.61 μm^2^ vs. 205.12 ± 8.77 μm^2^, respectively) (Figure 2C, Appendix A), as demonstrated by the higher distribution of the nuclear area values of 3T3 cells in the SMG group than the control group (Figure 2F,G).

### 3.3. SMG Induces Alteration in Cytoskeleton Distribution

Cell division requires changes in the cytoskeleton, especially microfilaments and microtubules. Fluorescent staining showed that 3T3 cells in the control group had a higher density of microfilament bundles than cells under SMG (Figure 3). Moreover, microtubule staining demonstrated that 3T3 cells in the control group had perinuclear accumulation of microtubules, while the microtubules surrounded the nucleus in 3T3 cells in the SMG group (Figure 4). Western blot analysis showed that the expression of β-actin and α-tubulin 3 was downregulated in 3T3 cells under SMG (Figure 5). In addition, microvilli were found to be abundant in control 3T3 cells, while 3T3 cells under SMG showed reduced microvilli (Figure 3). 

### 3.4. Cell Cycle Progression

Cell morphology had a close association with cell division and proliferation, as indicated by the cell cycle progression. The analysis by the cell cycle app showed that the cell percentage in the G0/G1 phase was lower for the control group than the SMG group (63.46 ± 2.13% vs. 66.14 ± 1.97%, respectively) (*p* < 0.001), while the cell percentage in the G2/M phase was higher for the control group than the SMG group (19.90 ± 1.01% vs. 16.45 ± 1.60%, respectively) (*p* < 0.001) (Figure 6A,B, Appendix A). These results indicate that induced SMG led 3T3 cells to the arrest phase in cell cycle progression. The decreased percentage of the G0/G1 phase for cells in the SMG group was found to be associated with the downregulation of Cdk4 and Cdk6 expression (Figure 6C). The arrest phase transition was correlated with the change in cell density. The cell density was lower for the SMG group (14.04 × 10^3^ cells/well) than the control group (14.97 × 10^3^ cells/well) (Figure 6D, Appendix A).

We also assessed the morphological changes in cell division structures, including contractile ring and spindle. The interactions between the mitotic spindle, the contractile ring, and the plasma membrane guarantee the correct placement of the cleavage furrow between chromosomes and the formation of new membrane compartments. As seen in Figure 7, the 3T3 cells in the control group exhibited a normal contractile ring during cell separation, whereas an asymmetrical form and division were noted in 3T3 cells under SMG. According to the results of microtubule staining, the 3T3 cells in the SMG and control groups displayed normal spindle formation during mitosis (Figure 7). A normal polar spindle microtubule structure was observed during cytokinesis in the control 3T3 cells, whereas the 3T3 cells under SMG showed an irregular form.

## 4. Discussion

Previous studies used mouse fibroblasts as a good model to assess the effects of microgravity or hypergravity on cell structure and function. The actin dynamics changed in mouse fibroblasts under SMG, demonstrated by downregulation of F-actin and G-actin, leading to extensive actin remodeling in these cells [17]. SMG could induce decreased apoptosis in mouse fetal fibroblasts [18]. Yang and colleagues (2016) showed that SMG could alter the circadian rhythm of 3T3 cells by influencing the mRNA expression of circadian genes and induce a change in Per1 and Per2 expression [19]. Fibroblasts exhibit high sensitivity to SMG, which validates the possible significance of the TGF-β1/Smad3 signaling pathway in regulating wound healing [20]. A recent study revealed that changes in fibroblast structures and delayed cell migration were the main effects of altered gravity, contributing to alterations in the fibroblasts’ function related to wound healing [21]. These studies indicated the gravisensitivity of mouse fibroblasts to SMG, suggesting that these cells are suitable for assessing the influence of microgravity on characteristics such as proliferation, function, and morphology.

The size and mass of organisms play important roles in adaptation and evolution under the influence of gravity [22]. Gravity was shown to have different effects on cells and their components, including the nucleus, cytoskeleton, membrane, and organelle [23,24,25,26]. Cell structure changes depend on the gravity conditions, such as hypergravity or microgravity. Moreover, changes in gravity also affect the expression of genes that regulate cell structure and function [27,28,29]. In this study, we found that SMG not only affected changes in cell morphology, nucleus, and cytoskeleton, but also altered the expression of structural and cell cycle related genes. In this study, we found that SMG induced increases in 3T3 cell size and nuclear area, resulting in changes in nuclear morphology and decreased nuclear shape. Nuclear reconstruction is essential for cell cycle progression, in which DNA replication and the synthesis of other cellular components are enhanced. In mitosis, cell division requires the chromatin condensation process, which enhances nuclear intensity [30]. In the current work, the intensity of Hoechst 33342-stained nuclei was assessed by the cell cycle app [30]. The 3T3 cells in the SMG group exhibited lower nuclear intensity than the control cells, correlating with the attenuation of nuclear condensation and reduction of cell division in 3T3 cells. 

The cytoskeleton plays an important role in constructing the cell shape and structures, supporting cell movements, and contributing to cell interactions [31]. Microfilaments and microtubules are the two main structural components of the cytoskeleton that are essential to cell division [32]. In mitosis, microtubules extend from the centrioles to the cell plate, forming spindles, which segregate chromosomes into daughter cells [33]. In addition, microfilaments contribute to the construction of cleavage furrows during mitosis [34]. Actin and tubulin are components of microfilaments and microtubules, respectively. Changes in their synthesis can alter the construction of microfilaments and microtubules. In the current investigation, 3T3 cells under SMG exhibited downregulated expression of β-actin and α-tubulin 3, leading to changes in the formation of spindles and cleavage furrows, which support cell division. 

As reported in a previous study, the surface and shape of cells change during mitosis [35]. A flat form with few microvilli is observed in cells from prophase to metaphase [36]. The number of microvilli in cells is remarkably increased in anaphase, and cells exhibit an apical shape [36]. However, after completing cell division, cells exhibit a flat form with a marked decrease in microvilli and enter the interphase [36]. In this work, a large number of 3T3 cells in the control group had an apical shape with abundant microvilli, while cells under SMG displayed a flat form with dramatically diminished microvilli. This indicates that SMG induced an attenuation of microvilli formation in 3T3 cells. The organization of microfilaments and microtubules is associated with CDK4 and CDK6 [37,38]. A recent study reported that the phosphorylation and enzymatic activity of SMYD2, an α-tubulin methyltransferase, are positively regulated by CDK4/6. Microtubule dynamics are modified by alterations in CDK4/6 and SMYD2 signaling, which impact microtubule stability [39]. The interaction of CKD6 with cytoskeleton-related proteins regulates this organization, especially the polymerization of actin [40]. Cdk6-deficient cells were found to attenuate the formation of F-actin, which is polymerized to form microfilaments [40]. In this study, 3T3 cells under SMG had reduced expression of CDK4 and CDK6, which correlates to the decrease in β-actin, leading to the diminished formation of microfilaments in 3T3 cytoplasm and reduced cell division. In conclusion, our results suggest that SMG induces morphological changes in 3T3 cells, as demonstrated by alterations in cytoskeleton distribution, cell and nuclear size, and cell cycle progression. This remodeling is generated by the downregulation of main cytoskeletal proteins and cell cycle related regulators.

Cytokinesis, the last step of cell division, requires the formation of contractile rings and spindles [41,42]. Contractile rings are formed in this final step. They are composed of filamentous actin (F-actin) and the motor protein myosin-2, along with additional structural and regulatory proteins [43]. Contractile rings produce the constricting force that divides a single cell into two cells during cell division [44]. The contractile ring is attached to the plasma membrane so that when it contracts, it forms a cleavage furrow that enhances cell division [45]. In cytokinesis, the central spindle performs a variety of functions, including directing the position of the cleavage furrow, transporting membrane vesicles to the cleavage furrow, and forming the midbody structure required for the last stages of division [41]. This study shows the alterations of the contractile ring and polar spindles in 3T3 cells under SMG, as demonstrated by the asymmetrical shape and irregular morphology of these structures. This result indicates that SMG impacts cytokinesis by inducing the deformation of division structures. 

The cell cycle is associated with a number of shape changes that depend on the exact alteration of a cell’s physical characteristics. During cell division, individual cells must maintain strict control over their shape. Mechanical characteristics and external forces combine to form the physical properties of a cell. Cells have undergone morphological changes to adapt to the natural gravity conditions of the Earth. This has resulted in the formation of cytoskeletons with specialized and functional structures, such as microfilaments and microtubules. In this study, the 3D clinostat system was used to investigate the effects of SMG on 3T3 cell morphology during cell cycle progression. Simulated microgravity conditions were created by continuous rotation of the two axes, which altered the force of gravity on the 3T3 cells. The external forces, one of the two factors that affect the physical properties of cells, were changed. Thus, the attenuated microgravity produced by the 3D clinostat simulator had significant initial effects on the structure of the cytoskeleton, specifically the redistribution of microfilaments and microtubules. This led to subsequent changes in cell morphology, such as increased cell and nuclear size, decreased nuclear shape value, and irregular formation of the contractile ring and polar spindle during cytokinesis.

## 5. Conclusions

This study shows that SMG induces morphological changes in 3T3 cells by inducing changes in the distribution of microfilaments and microtubules. The downregulation of the main cytoskeletal proteins, including β-actin and α-tubulin 3, was shown to contribute to the modification of these structures. In addition, SMG also causes changes in cell cycle progression of 3T3 cells. 

## Figures and Tables

**Figure 1 cells-13-00344-f001:**
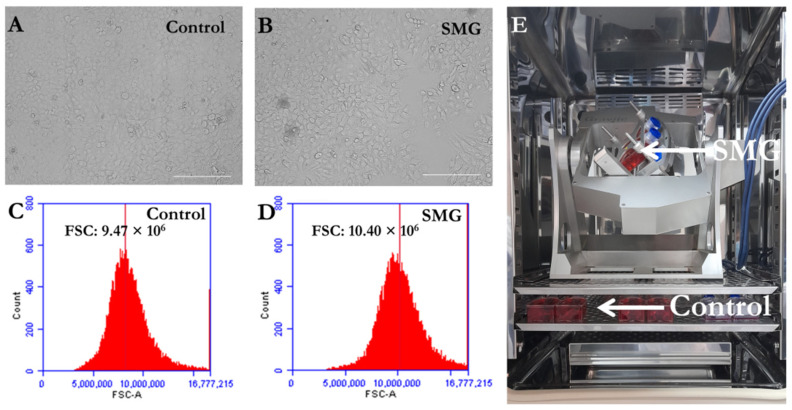
Proliferation of 3T3 cells in control and SMG groups. (**A**,**B**) Cell morphology of 3T3 cells in control and SMG groups. (**C**,**D**) FCS values for control and SMG groups (*n* = 5). (**E**) Gravite^®^ operation in CO_2_ incubator. Scale bar = 223.64 µm.

**Figure 2 cells-13-00344-f002:**
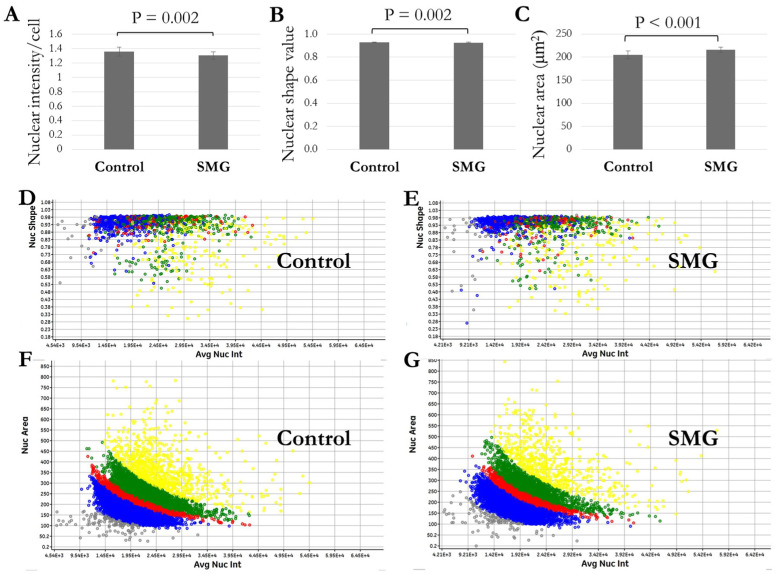
Analysis of 3T3 nuclear morphology. (**A**) Nuclear intensity value/cell (*n* = 24). (**B**) Nuclear shape value (*n* = 24). (**C**) Nuclear area (*n* = 24). (**D**,**E**) Distribution of 3T3 nuclear shape values relative to nuclear intensity. (**F**,**G**) Distribution of 3T3 nuclear area values relative to nuclear intensity. Gray indicates percentage of nuclei < 2n, blue indicates percentage of nuclei in G0/G1 phase, red indicates percentage of nuclei in S phase, green indicates percentage of nuclei in G2/M phase, and yellow indicates percentage of nuclei > 4n.

**Figure 3 cells-13-00344-f003:**
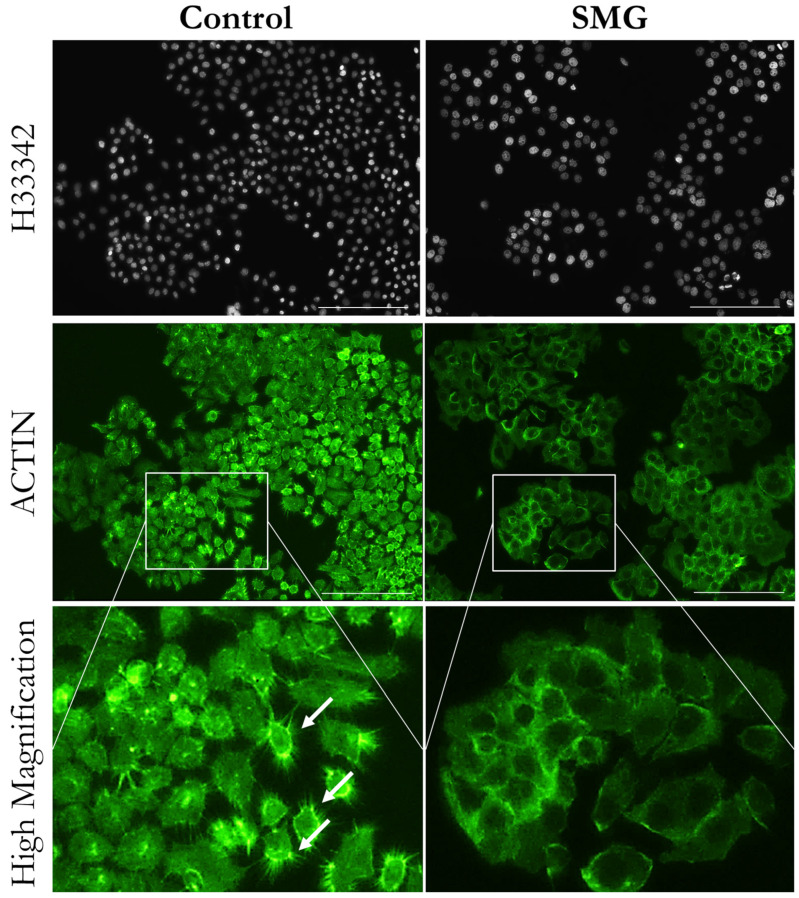
Distribution of microfilament bundles of 3T3 cells. Microfilaments were stained with phalloidin (green color), and nuclei were counterstained with H33342. White arrows indicate microvilli. Scale bar = 223.64 µm.

**Figure 4 cells-13-00344-f004:**
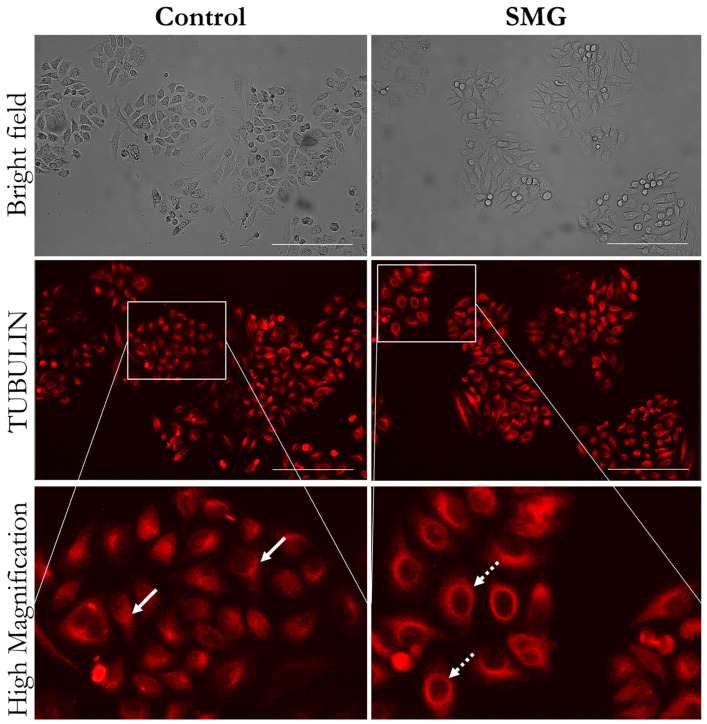
Distribution of microtubules of 3T3 cells. Microtubules were stained with SiR-tubulin (red color). White arrows indicate perinuclear accumulations of microtubules; dashed arrows indicate distribution of microtubules surrounding nucleus. Scale bar = 223.64 µm.

**Figure 5 cells-13-00344-f005:**
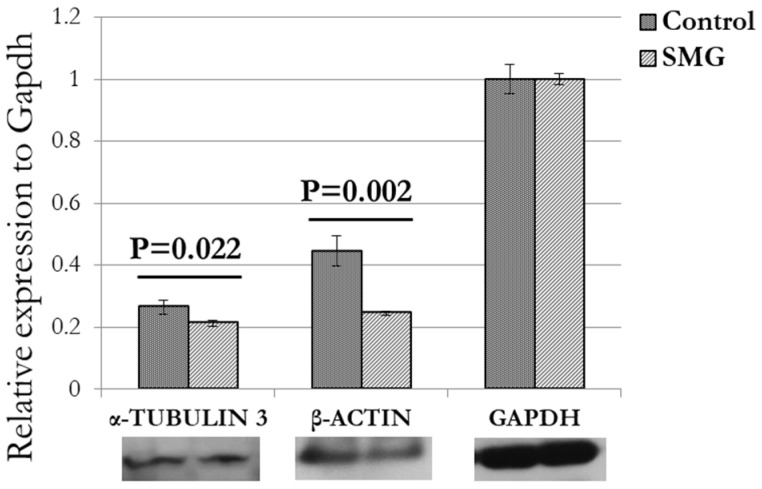
Western blot analysis of major structural proteins in 3T3 cells. α-Tubulin 3 and β-actin were downregulated in 3T3 cells under SMG (*n* = 3). GAPDH was used as internal control.

**Figure 6 cells-13-00344-f006:**
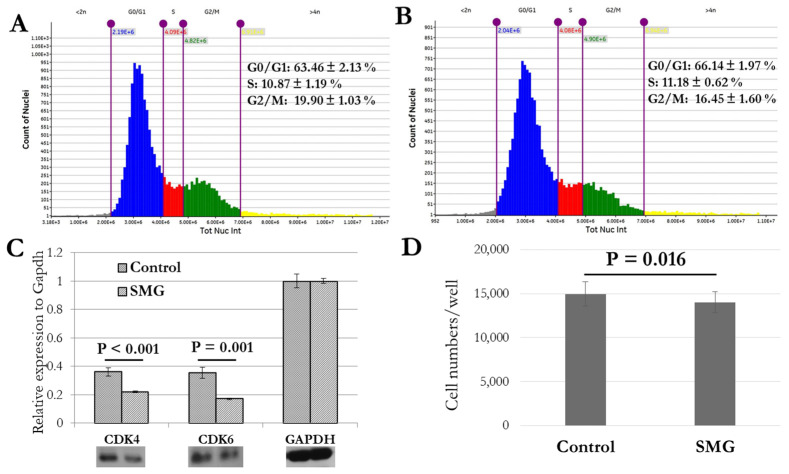
Cell cycle progression analysis. (**A**,**B**) Cell cycle of 3T3 cells in control and SMG groups was analyzed by cell cycle app of Cytell microscope (*n* = 24). Gray indicates percentage of nuclei < 2n, blue indicates percentage of nuclei in G0/G1 phase, red indicates percentage of nuclei in S phase, green indicates percentage of nuclei in G2/M phase, and yellow indicates percentage of nuclei > 4n. (**C**) Western blot analysis of major cell cycle-related proteins in 3T3 cells (*n* = 3). (**D**) Number of 3T3 cells was counted by cell cycle app (*n* = 24).

**Figure 7 cells-13-00344-f007:**
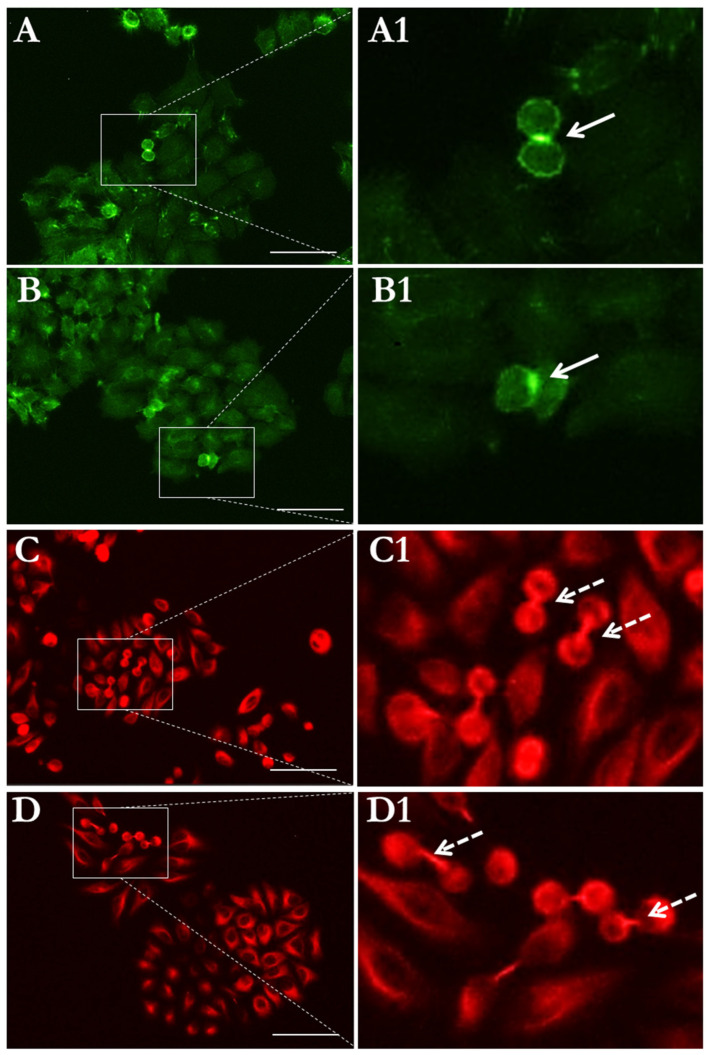
Morphology of cell division related structures in 3T3 cells: (**A**,**A1**) contractile ring in 3T3 cells in control group; (**B**,**B1**) contractile ring in 3T3 cells in SMG group; (**C**,**C1**) polar spindle in 3T3 cells in control group; (**D**,**D1**) polar spindle in 3T3 cells in SMG group. Microfilaments were counterstained using phalloidin (green), and microtubules were stained with SiR-tubulin (red). White arrows indicate contractile rings, and dashed arrows indicate polar spindles. Scale bar = 100 µm.

## Data Availability

Data are contained within the article and Appendix A.

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
