# Peer review of "Morphological Changes of 3T3 Cells under Simulated Microgravity"

_cells, 2024, doi:10.3390/cells13040344_

Round 1
Reviewer 1 Report
Comments and Suggestions for Authors
The article “The morphological changes of 3T3 cells under simulated microgravity”, opportunities by Minh Thi Tran et al. is very interesting.
The authors to present the results of simulated microgravity (SMG) applied 3D clinostat and assess the effects of SMG on cell morphology. This study shows that SMG induces morphological changes in 3T3 cells by inducing changes in nuclear and cell size, altering the distribution of microfilaments and microtubules.
However, several points have to be addressed:
Line 57: Please full expand abbreviation: SMG...
Results
Line 238: Please indicate (P = …)
1. As in the previous study (Int. J. Mol. Sci. 2021), the authors again present the results of the experiment at a one point time (72 h) ... but do not explain why... Please explain the necessity of the time selected.
2. The number of independent experiments should be indicated in each Figure legend.
3. Please full expand all abbreviation in special section: 3T3 cell, LDS, TBST, HRP …
Author Response
Dear Prof. Dr. Cord Brakebusch, Editorial Board and Reviewers,
We are very grateful to the Editor for your consideration of our manuscript. We would like to thank the Reviewers for careful and thorough reading of the manuscript and for the thoughtful comments and constructive suggestions, which help to improve the quality of this manuscript. Each comment has been carefully considered point by point and responded. Responses to the reviewers and changes in the revised manuscript are as follows.
Reviewer 1:
- Line 57: Please full expand abbreviation: SMG...
Author's response:
Thank you for your comment, we have added the abbreviation in our manuscript
- Line 238: Please indicate (P = …)
Author's response:.
We appreciate your comment. The format of P value was corrected to “P = …” in Figure 6E at line 238.
- As in the previous study (Int. J. Mol. Sci. 2021), the authors again present the results of the experiment at a one point time (72 h) ... but do not explain why... Please explain the necessity of the time selected.
Author's response:
Thank you for your comment. Humans and cells are very sensitive to changes in gravity. About half of the astronauts endure "space adaptation syndrome" for one to three days after entering microgravity (Connors et al., 1985; Merz and Beverly, 1986). Many studies show that cells have many changes in a short time (2-4 days) when exposed to simulated microgravity (Damm et al., 2013; Yan et al., 2015; Tan et al., 2018; Touchstone et al., 2019). Evaluation of cell proliferation and cytoskeletal structure induced by simulated microgravity in short-term could clarify the alteration and the adaptation of cells under simulated microgravity. Thus, this study was designed to estimate the morphological changes of 3T3 cells under simulated microgravity at 72 h.
References:
Connors, Mary M.; Harrison, Albert A.; Akins, Faren R. (1985). Living Aloft: Human Requirements for Extended Spaceflight (NASA SP-483, p. 35-51). NASA Scientific and Technical Information Branch.
Merz, Beverly (1986 October 17). The Body Pays a Penalty for Defying the Law of Gravity. In, Journal of the American Medical Association (vol. 256, no. 15, p. 2040-2041). American Medical Association.
Touchstone H, Bryd R, Loisate S, Thompson M, Kim S, Puranam K, Senthilnathan AN, et al. Recovery of stem cell proliferation by low intensity vibration under simulated microgravity requires LINC complex. NPJ Microgravity 2019; 5, 11.
Yan M, Wang Y, Yang M, Liu Y, Qu B, Ye Z, Liang W, et al. The effects and mechanisms of clinorotation on proliferation and differentiation in bone marrow mesenchymal stem cells. Biochem Biophys Res Commun 2015; 460: 327-332.
Tan X, Xu A, Zhao T, Zhao Q, Zhang J, Fan C, Deng Y, et al. Simulated microgravity inhibits cell focal adhesions leading to reduced melanoma cell proliferation and metastasis via FAK/RhoA-regulated mTORC1 and AMPK pathways. Sci Rep 2018; 8, 3769.
Damm TB, Richard S, Tanner S, Wyss F, Egli M, Franco-Obregon A. Calcium-dependent deceleration of the cell cycle in muscle cells by simulated microgravity. FASEB J 2013; 27: 2045-2054.
- The number of independent experiments should be indicated in each Figure legend.
Author's response:
The number of independent experiments was indicated in each Figure legend.
- Please full expand all abbreviation in special section: 3T3 cell, LDS, TBST, HRP …
Author's response:
The abbreviation is added to manuscript
We hope that our corrections could meet your requirements,
Thank you so much.
Reviewer 2 Report
Comments and Suggestions for Authors
This study investigated the structure alterations of cytoskeleton and cell cycle progression of 3T3 cells under simulated microgravity. However, the relationship between cytoskeleton structure changes and cell cycle progression that the author illustrated is lack of direct evidence. Hope that they can supplement new experiments to prove it. For instance, if the cytoskeleton were depolymerize, how the cell cycle progression changes? Below are detailed suggestions:
Methods:
1. Please explain why the author chose Mode C but not Mode B or A? What’s the difference among these modes?
2. Why the author chose 72h for simulated microgravity?
3. It seems the author used the culture flasks to rotate cells (Figure 1E), why the author described they used 96-well plate?
4. What are 3T3 cells? The mouse embryo fibroblasts? If so, why do the cell morphology in Figures 3-5 appear more round and lack the typical fusiform shape even in the control groups?
Results:
1. The author claimed that 3T3 exhibited a spindle-like appearance, while in the fluorescent images of microfilament bundles in Figure 3, the cells were cultured too crowded to see the typical structure. I recommend to adjust the incubated density to see if they appear the same morphology.
Discussion:
1. The author described the microvilli of actin filaments in cells, and consider the decrease of the microvilli in SMG group revealed the reduction of cell division. This judgment is arbitrary. How the author demonstrates this phenomenon is related to the cell division?
2. What is the relationship between CDK4 and actin filaments or microtubules? What is the function of it?
3. Line344, what dose the author mean by “a decrease in nuclear morphology?”
Conclusion:
The author should clean up the logical relations between cytoskeleton changes and the size of cell and nuclear size. Do the former affect the later or they both are affected by the simulated microgravity? The conclusion now is inconsistent with the last part in discussion.
Comments on the Quality of English LanguageIt should be ok and can be improved.
Author Response
Dear Prof. Dr. Cord Brakebusch, Editorial Board and Reviewers,
We are very grateful to the Editor for your consideration of our manuscript. We would like to thank the Reviewers for careful and thorough reading of manuscript and for the thoughtful comments and constructive suggestions, which help to improve the quality of this manuscript. Each comment has been carefully considered point by point and responded. Responses to the reviewers and changes in the revised manuscript are as follows.
Methods:
- Please explain why the author chose Mode C but not Mode B or A? What’s the difference among these modes?
Author's response:
As presenting in Materials and Methods, there are four microgravity simulation programs: mode A (×4 rpm), mode B (×3 rpm), mode C (×2 rpm), and mode D (×1 rpm). The difference between the programs is the rotation speed of the 2 frames of Gravite machine. The program for experiments on animal cells given by the manufacturer is mode C. Therefore, we apply this mode for experiments on 3T3 cells in this study.
- Why the author chose 72h for simulated microgravity?
Author's response:
Humans and cells are very sensitive to changes in gravity. About half of the astronauts endure "space adaptation syndrome" for one to three days after entering microgravity (Connors et al., 1985; Merz and Beverly, 1986). Many studies show that cells have many changes in a short time (2-4 days) when exposed to simulated microgravity (Damm et al., 2013; Yan et al., 2015; Tan et al., 2018; Touchstone et al., 2019). Evaluation of cell proliferation and cytoskeletal structure induced by simulated microgravity in short-term could clarify the alteration and the adaptation of cells under simulated microgravity. Thus, this study was designed to estimate the morphological changes of 3T3 cells under simulated microgravity at 72 h.
References:
Connors, Mary M.; Harrison, Albert A.; Akins, Faren R. (1985). Living Aloft: Human Requirements for Extended Spaceflight (NASA SP-483, p. 35-51). NASA Scientific and Technical Information Branch.
Merz, Beverly (1986 October 17). The Body Pays a Penalty for Defying the Law of Gravity. In, Journal of the American Medical Association (vol. 256, no. 15, p. 2040-2041). American Medical Association.
Touchstone H, Bryd R, Loisate S, Thompson M, Kim S, Puranam K, Senthilnathan AN, et al. Recovery of stem cell proliferation by low intensity vibration under simulated microgravity requires LINC complex. NPJ Microgravity 2019; 5, 11.
Yan M, Wang Y, Yang M, Liu Y, Qu B, Ye Z, Liang W, et al. The effects and mechanisms of clinorotation on proliferation and differentiation in bone marrow mesenchymal stem cells. Biochem Biophys Res Commun 2015; 460: 327-332.
Tan X, Xu A, Zhao T, Zhao Q, Zhang J, Fan C, Deng Y, et al. Simulated microgravity inhibits cell focal adhesions leading to reduced melanoma cell proliferation and metastasis via FAK/RhoA-regulated mTORC1 and AMPK pathways. Sci Rep 2018; 8, 3769.
Damm TB, Richard S, Tanner S, Wyss F, Egli M, Franco-Obregon A. Calcium-dependent deceleration of the cell cycle in muscle cells by simulated microgravity. FASEB J 2013; 27: 2045-2054.
- It seems the author used the culture flasks to rotate cells (Figure 1E), why the author described they used 96-well plate?
Author's response:
This study applied the culture flasks and 96-well plate for cell culture. The culture flasks were used for cell culture in analysis of cell cycle progression by flow cytometry and evaluation of protein expression by western blot. The other experiments applied 96-well plates for cell culture, including cell density analysis, nuclear morphology, microfilament staining, and microtubule staining, which were analyzed by Cytell microscope. Thus, 96-well plates were used for cell culture and were fixed in the stage of Gravity Controller Gravite.
- What are 3T3 cells? The mouse embryo fibroblasts? If so, why do the cell morphology in Figures 3-5 appear more round and lack the typical fusiform shape even in the control groups?
Author's response:
3T3 is a fibroblast cell line that was isolated from a mouse NIH/Swiss embryo. This study found that SMG resulted in the morphogical changes and the decrease of proliferation in 3T3 cells. SMG induced the nuclear-surrounded distribution of microtubule, which cells more round lack the typical fusiform shape. However, perinuclear accumulation of microtubule was determined in cells from control group. 3T3 cells from control group also exhibited the more round shape because they were in cell division process, demonstraing by the appearance of round and apical shape with the abundance of microvilli, while 3T3 cells under SMG condition displayed the flat form with dramatically diminishing microvilli.
Results:
- The author claimed that 3T3 exhibited a spindle-like appearance, while in the fluorescent images of microfilament bundles in Figure 3, the cells were cultured too crowded to see the typical structure. I recommend to adjust the incubated density to see if they appear the same morphology.
Author's response:
Thank you so much for your comment. 3T3 cells were seeded at a density of 3 × 103 cells/well. After 72 h, cell density reached to about 1.5 × 104 cells/well. The high density cause to changes in spindle-like shape in 3T3 cells. Thus, we have corrected “3T3 cells in both groups exhibited their normal morphologies, including single attachment and a spindle-like appearance” to “3T3 cells in both groups exhibited their normal morphologies” in the fisrt paragragh of the Results.
Discussion:
- The author described the microvilli of actin filaments in cells, and consider the decrease of the microvilli in SMG group revealed the reduction of cell division. This judgment is arbitrary. How the author demonstrates this phenomenon is related to the cell division?
Author's response:
Thank you for your comment, we have corrected the evaluation of this result. The sentence “This revealed that SMG induced a attenuation of microvilli formation in 3T3 cells” was corrected to “This revealed that SMG induced an attenuation of microvilli formation in 3T3 cells”.
- What is the relationship between CDK4 and actin filaments or microtubules? What is the function of it?
Author's response:
Relationship between CDK4 and microtubules, and their function are presented in Discussion.
- Line344, what dose the author mean by “a decrease in nuclear morphology?”
Author's response:
A phrase “a decrease in nuclear morphology” was corrected to “a decrease in nuclear shape value”.
Conclusion:
- The author should clean up the logical relations between cytoskeleton changes and the size of cell and nuclear size. Do the former affect the later or they both are affected by the simulated microgravity? The conclusion now is inconsistent with the last part in discussion.
Author's response:
In this study, we have recorded that the effects of SMG on changes in cytoskeleton, cells size, and nuclear size. Several study reported that SMG results in cytoskeletal reorganization, which leading to cell size changes. This alteration was also observed in 3T3 cells. We have corrected the conclsion.
- Comments on the Quality of English Language, It should be ok and can be improved.
Author's response:
The manuscript has been corrected to improve the quality of English Language.
We hope that our corrections could meet your requirements,
Thank you so much.
Reviewer 3 Report
Comments and Suggestions for Authors
Lack of gravity is a major risk factor during space flight. Ground-based simulation of microgravity effects is a very potential tool for elucidating the microgravity effects at the cellular and molecular levels. Currently, the simulation of microgravity effects using various devices such as Gravite is the most widely used experimental approach for cell experiments. The different types of cells have been studied in detail.
The main concern with this study arises from these already well done studies. The reviewer did not see any obvious novelty compared to what has already been published. The authors need to specify why this work was necessary and what exactly they have discovered that is new.
Specific comments.
- Why were 3T3 cells used for the experiments? This is a well-studied line of murine fibroblasts. Why is the gravisensitivity of these cells of interest for space cell biology?
- Why was 72 h exposure chosen?
- For the SMG exposure, 96-well plates covered with parafilm were used. Were air bubbles present in the medium in the wells and was medium leakage observed?
- The flow cytometry section is sparsely and incomprehensibly written. What method was used to remove cells from the culture tubes? Why were cells fixed with 4% paraformaldehyde for cell cycle analysis? Usually alcohol fixation is used. What protocol was used for cell cycle analysis? How many cells were analyzed by flow cytometry?
- Standard deviations for FSC values are not reported.
- How many cells were analyzed to characterize changes in nuclear shape and size?
- Figure 2. The authors claim significant differences in A, B, and C. Despite the claimed statistical significance, the changes often do not exceed 5%. These differences are so small. Please explain.
- Were the density of microfilaments and the number of cells with microvilli quantified? Are they really microvilli or are they still stress fibrils?
- Fig. 4 shows photographs showing the distribution of microtubules. It can be seen that under both static and experimental conditions, cells are observed with both "voids" around the nuclei and cells with more uniform tubulin filling. Have you quantified and compared the number of cells with and without voids under static and experimental conditions?
- For more accurate identification of structures associated with cell division, it is useful to use chromosome stain (Hoechst 3342 or DAPI) along with tubulin to identify division spindles.
- Line 301. It would be good to give quantitative data on stress fibrils and microvilli.
- Due to the lack of description of the flow cytometry method and the cell cycle assessment protocol, it is not possible to evaluate the cell cycle data obtained in this paper.
In its present form, the discussion is overloaded with details and speculations. It needs to focus more on the novelty of the data obtained.In its present form, the discussion is overloaded with details and speculations. It needs to focus more on the novelty of the data obtained.
Author Response
Dear Prof. Dr. Cord Brakebusch, Editorial Board and Reviewers,
We are very grateful to the Editor for your consideration of our manuscript. We would like to thank the Reviewers for careful and thorough reading of manuscript and for the thoughtful comments and constructive suggestions, which help to improve the quality of this manuscript. Each comment has been carefully considered point by point and responded. Responses to the reviewers and changes in the revised manuscript are as follows.
- Why were 3T3 cells used for the experiments? This is a well-studied line of murine fibroblasts. Why is the gravisensitivity of these cells of interest for space cell biology?
Author's response:
Thank you for your comments. The previous studies have used mouse fibroblast as a good model to assess the effects of microgravity or hypergravity on several cell structure and function. The actin dynamics was changed in mouse fibroblasts under SMG condition, demonstrated by a down-regulation of F-actin and G-actin, leading to the extensive actin remodeling in these cells (Moes et al., 2007). SMG condition could induce a decrease apoptosis in mice fetal fibroblasts (Beck et al., 2012). Yang and his colleagues (2016) showed that SMG could altered the circadian rhythm of 3T3 cells by influence the mRNA expression of circadian genes and induced a change in Per1 and Per2 expression. Fibroblasts exhibit high sensitivity to SMG, which has clarified the possible significance of the TGF-β1/Smad3 signaling pathway in regulating wound healing (Zhou et al., 2023). The recent study revealed that changes in fibroblasts structures and a delay of cell migration were the main effects of altered gravity, that contribute to alterations on the fibroblasts’ function related to wound healing (Radstake et al., 2023). These studies indicated the gravisensitivity of mouse fibroblasts to SMG, suggesting that these cells are suitable to assess the influence of microgravity on their characteristics such as proliferation, function, and morphology.
References:
Moes, M.J.A., Bijvelt, J.J. & Boonstra, J. Actin dynamics in mouse fibroblasts in microgravity. Microgravity Sci. Technol 19, 180–183 (2007).
Beck, M., Tabury, K., Moreels, M., Jacquet, P., Van Oostveldt, P., De Vos, W.H., & Baatout, S. (2012). Simulated microgravity decreases apoptosis in fetal fibroblasts. International Journal of Molecular Medicine, 30, 309-313.
Shuhong Yang, Yanyou Liu, Yunyun Yang, Zhenhua Yang, Shuting Cheng, Wang Hou, Yuhui Wang, Zhou Jiang, Jing Xiao, Huiling Guo & Zhengrong Wang (2016) Simulated microgravity influences circadian rhythm of NIH3T3 cells, Biological Rhythm Research, 47:6, 897-907.
Yuhao Zhou, Wenjun Lv, Xiufen Peng, Yansiwei Cheng, Yun Tu, Guanbin Song, Qing Luo, Simulated microgravity attenuates skin wound healing by inhibiting dermal fibroblast migration via F‐actin/YAP signaling pathway, Journal of Cellular Physiology, 10.1002/jcp.31126, 238, 12, (2751-2764), (2023).
Radstake, W.E., Gautam, K., Miranda, S. et al. Gravitational effects on fibroblasts’ function in relation to wound healing. npj Microgravity 9, 48 (2023)
- Why was 72 h exposure chosen?
Author's response:
Thank you for your comment. Humans and cells are very sensitive to changes in gravity. About half of the astronauts endure "space adaptation syndrome" for one to three days after entering microgravity (Connors et al., 1985; Merz and Beverly, 1986). Many studies show that cells have many changes in a short time (2-4 days) when exposed to simulated microgravity (Damm et al., 2013; Yan et al., 2015; Tan et al., 2018; Touchstone et al., 2019). Evaluation of cell proliferation and cytoskeletal structure induced by simulated microgravity in short-term could clarify the alteration and the adaptation of cells under simulated microgravity. Thus, this study was designed to estimate the morphological changes of 3T3 cells under simulated microgravity at 72 h.
References:
Connors, Mary M.; Harrison, Albert A.; Akins, Faren R. (1985). Living Aloft: Human Requirements for Extended Spaceflight (NASA SP-483, p. 35-51). NASA Scientific and Technical Information Branch.
Merz, Beverly (1986 October 17). The Body Pays a Penalty for Defying the Law of Gravity. In, Journal of the American Medical Association (vol. 256, no. 15, p. 2040-2041). American Medical Association.
Touchstone H, Bryd R, Loisate S, Thompson M, Kim S, Puranam K, Senthilnathan AN, et al. Recovery of stem cell proliferation by low intensity vibration under simulated microgravity requires LINC complex. NPJ Microgravity 2019; 5, 11.
Yan M, Wang Y, Yang M, Liu Y, Qu B, Ye Z, Liang W, et al. The effects and mechanisms of clinorotation on proliferation and differentiation in bone marrow mesenchymal stem cells. Biochem Biophys Res Commun 2015; 460: 327-332.
Tan X, Xu A, Zhao T, Zhao Q, Zhang J, Fan C, Deng Y, et al. Simulated microgravity inhibits cell focal adhesions leading to reduced melanoma cell proliferation and metastasis via FAK/RhoA-regulated mTORC1 and AMPK pathways. Sci Rep 2018; 8, 3769.
Damm TB, Richard S, Tanner S, Wyss F, Egli M, Franco-Obregon A. Calcium-dependent deceleration of the cell cycle in muscle cells by simulated microgravity. FASEB J 2013; 27: 2045-2054.
- For the SMG exposure, 96-well plates covered with parafilm were used. Were air bubbles present in the medium in the wells and was medium leakage observed?
Author's response:
During SMG exposure, there was no air bubbles present in the medium in the wells and medium leakage was not observed.
- The flow cytometry section is sparsely and incomprehensibly written. What method was used to remove cells from the culture tubes? Why were cells fixed with 4% paraformaldehyde for cell cycle analysis? Usually alcohol fixation is used. What protocol was used for cell cycle analysis? How many cells were analyzed by flow cytometry?
Author's response:
After 72 h, the culture medium was discarded and cells were washed twice by 4 ml PBS. Cell detachment was performed with 1 ml trypsin-EDTA 0.25% (TRY-2B, Capricorn Scientific, Ebsdorfergrund, Germany) for 3 min. 3 ml cell culture medium was added to flask and mixed well. Cell suspension was transferred to 15 ml tube and centrifuged at 1500 rpm for 5 min. The medium was discarded and cell pellet was resuspended with 4 ml PBS. Cell suspension was applied for centrifuging at 1500 rpm for 5 min. The medium was discarded and the pellet was resuspended with 4% paraformaldehyde solution (09154-85, Nacalai Tesque, Kyoto, Japan) for 30 min to fix cells. Cell pellet was washed twice by 4 ml PBS and Cell nuclear were then stained with 5 µL propidium iodide (51-66211E, BD Biosciences, San Jose, CA, United States). The cell cycle progression was subsequently analyzed using a flow cytometer BD Accuri C6 Plus (BD Biosciences, San Jose, CA, United States). However,
- Standard deviations for FSC values are not reported.
Author's response:
The SD for FSC values were presented in the results and Figure 1.
- How many cells were analyzed to characterize changes in nuclear shape and size?
Author's response:
The cell number to characterize changes in nuclear shape and size were present in supplementary data.
- Figure 2. The authors claim significant differences in A, B, and C. Despite the claimed statistical significance, the changes often do not exceed 5%. These differences are so small. Please explain.
Author's response:
In this study, we found that although SMG induces changes in some cell and nuclear morphological characteristics, the changes often do not exceed 5% as your comment. A limitation of this study is that only one time point (72 h) was chosen for survey. In the next study, we will investigate many different time points to clarify the effect of SMG on the morphological characteristics of 3T3 cell nuclei.
- Were the density of microfilaments and the number of cells with microvilli quantified? Are they really microvilli or are they still stress fibrils?
Author's response:
Thank you for your comment. These strutures are microvilli. The figures of microfilament staining are 2D, thus we just characterize the microvilli appearance in 3T3 cells. To quantify density of microfilaments and the number of cells with microvilli, the higher resolution microscope is required to analyze and estimate the number of microvilli.
- Fig. 4 shows photographs showing the distribution of microtubules. It can be seen that under both static and experimental conditions, cells are observed with both "voids" around the nuclei and cells with more uniform tubulin filling. Have you quantified and compared the number of cells with and without voids under static and experimental conditions?
Author's response:
We have quantified and compared the number of cells with and without voids between control group and SMG group. The results showed that SMG showed 27.38 ± 6.20 % cells exposing tubulin bundle distribution around the nuclear, while the control group exhibited 7.82 ± 1.54. This result indicated that SMG condition induced a redistribution of microtubules in 3T3 cells.
- For more accurate identification of structures associated with cell division, it is useful to use chromosome stain (Hoechst 3342 or DAPI) along with tubulin to identify division spindles.
Author's response:
To stain microtubules, cells were directly incubated with Sir-Tubulin during SMG induction for 72 h (according to staining protocol of the manufacturer). After finishing the SMG induction, cells were taken out from Gravite and were immediately observed under fluorescent microscope. For nuclear staining, cells are treated with paraformaldehyde for fixing and permeabilized with trixton X100. However, these reagents cause the Sir-Tubulin degradation. Therefore, nuclear staining could not be applied to tubulin staining in this case. In the next study, we would like to use other reagents for tubulin staining, which will not be affected by nuclear staining reagents.
- Line 301. It would be good to give quantitative data on stress fibrils and microvilli.
Author's response:
thank you so much for your advice in assessing cytoskeletal structural changes. In this study, we only focus on describing the morphological changes of 3t3 cells under SMG conditions as well as the changes of some structures related to the cell division process. The quantitative data on stress fibers and microvilli require a higher resolution microscope such as confocal microscope or super resolution microscope. Therefore, in the next study, we will pay more attention to this issue to clarify the changes in cytoskeletal structures in cells under SMG conditions.
- Due to the lack of description of the flow cytometry method and the cell cycle assessment protocol, it is not possible to evaluate the cell cycle data obtained in this paper.
Author's response:
Thank you for your comment, the results of cell cycle analysis by flow cytometry were removed. We just keep cell cycle analysis by Cell Cycle App. of Cytell.
- In its present form, the discussion is overloaded with details and speculations. It needs to focus more on the novelty of the data obtained.In its present form, the discussion is overloaded with details and speculations. It needs to focus more on the novelty of the data obtained.
Author's response:
Thank you for your comment, we have revised the discussion and focused on the novelty of this study.
We hope that our corrections could meet your requirements,
Thank you so much.
Reviewer 4 Report
Comments and Suggestions for Authors
The manuscript by Minh Thi Tran et al entitled “The morphological changes of 3T3 cells under simulated microgravity” shows that SMG generated morphological changes in 3T3 cells by remodeling cytoskeleton structure and down-regulating 38 major structural protein and cell cycle regulators.
The study is carefully performed using robust methodology with abundant replicates. The statistics are valid and appropriate.
Major suggestions:
The manuscript would be greatly strengthened by placing it in context of prior studies. The discussion is currently largely a repetition of the results. Suggest discuss:
1. How do these results compare to the effects other stimuli to 3T3 cells in other studies?
2. How do 3T3 cells compare to other cell types in real or simulated microgravity?
3. What do earlier studies on 3T3 cell in real or simulated microgravity tell us regarding the current data set?
Minor issues:
2.1. Cell culture and SMG induction Suggest state fluid density and viscosity so readers can calculate the forces if desired.
2.3 Suggest state the final concentration of propidium iodide rather than volume added or state the total volume, so concentration can be deduced.
Author Response
Dear Prof. Dr. Cord Brakebusch, Editorial Board and Reviewers,
We are very grateful to the Editor for your consideration of our manuscript. We would like to thank the Reviewers for careful and thorough reading of manuscript and for the thoughtful comments and constructive suggestions, which help to improve the quality of this manuscript. Each comment has been carefully considered point by point and responded. Responses to the reviewers and changes in the revised manuscript are as follows.
- How do these results compare to the effects other stimuli to 3T3 cells in other studies?
Author's response:
Thank you so much for your comment. There are several studies in investigating the effects of gravity on 3T3 cells. SMG was able to modify the circadian rhythm of 3T3 cells, by changing the mRNA expression of circadian genes and inducing a shift in the expression of Per1 and Per2 (Yang et al., 2016). The hypergravity condition showed an effect on cell traction forces of 3T3 fibroblast, and induced the significant rearrangement of the actin cytoskeleton and the decrease in actin stress fibers at low hypergravity levels (Eckert et al., 2021). Our study found that SMG resutled in a reduction of actin expression and altered the distribution of actin filaments in 3T3 cells. These results revealed that 3T3 cells are sensitive to changes in gravity, including hypergravity and microgravity.
References:
Shuhong Yang, Yanyou Liu, Yunyun Yang, Zhenhua Yang, Shuting Cheng, Wang Hou, Yuhui Wang, Zhou Jiang, Jing Xiao, Huiling Guo & Zhengrong Wang (2016) Simulated microgravity influences circadian rhythm of NIH3T3 cells, Biological Rhythm Research, 47:6, 897-907.
Eckert J, van Loon JJWA, Eng LM, Schmidt T. Hypergravity affects cell traction forces of fibroblasts. Biophys J. 2021 Mar 2;120(5):773-780.
- How do 3T3 cells compare to other cell types in real or simulated microgravity?
Author's response:
As stated in the introduction, the changes in cell morphology and phenotypic transition was observed in many cells under SMG conditions. These alterations are generated by reorganizaition of cytoskeleton. In this study, 3T3 cells also exhibited the changes in morphology, demonstrated by the alteration of microfilament and microtubule.
- What do earlier studies on 3T3 cell in real or simulated microgravity tell us regarding the current data set?
Author's response:
Thank you for your comments. We have discuss more about the effects of SMG on 3T3 cells on the discussion.
The previous studies have used mouse fibroblast as a good model to assess the effects of microgravity or hypergravity on several cell structure and function. The actin dynamics was changed in mouse fibroblasts under SMG condition, demonstrated by a down-regulation of F-actin and G-actin, leading to the extensive actin remodeling in these cells (Moes et al., 2007). SMG condition could induce a decrease apoptosis in mice fetal fibroblasts (Beck et al., 2012). Yang and his colleagues (2016) showed that SMG could altered the circadian rhythm of 3T3 cells by influence the mRNA expression of circadian genes and induced a change in Per1 and Per2 expression. Fibroblasts exhibit high sensitivity to SMG, which has clarified the possible significance of the TGF-β1/Smad3 signaling pathway in regulating wound healing (Zhou et al., 2023). The recent study revealed that changes in fibroblasts structures and a delay of cell migration were the main effects of altered gravity, that contribute to alterations on the fibroblasts’ function related to wound healing (Radstake et al., 2023). These studies indicated the gravisensitivity of mouse fibroblasts to SMG, suggesting that these cells are suitable to assess the influence of microgravity on their characteristics such as proliferation, function, and morphology.
References:
Moes, M.J.A., Bijvelt, J.J. & Boonstra, J. Actin dynamics in mouse fibroblasts in microgravity. Microgravity Sci. Technol 19, 180–183 (2007).
Beck, M., Tabury, K., Moreels, M., Jacquet, P., Van Oostveldt, P., De Vos, W.H., & Baatout, S. (2012). Simulated microgravity decreases apoptosis in fetal fibroblasts. International Journal of Molecular Medicine, 30, 309-313.
Shuhong Yang, Yanyou Liu, Yunyun Yang, Zhenhua Yang, Shuting Cheng, Wang Hou, Yuhui Wang, Zhou Jiang, Jing Xiao, Huiling Guo & Zhengrong Wang (2016) Simulated microgravity influences circadian rhythm of NIH3T3 cells, Biological Rhythm Research, 47:6, 897-907.
Yuhao Zhou, Wenjun Lv, Xiufen Peng, Yansiwei Cheng, Yun Tu, Guanbin Song, Qing Luo, Simulated microgravity attenuates skin wound healing by inhibiting dermal fibroblast migration via F‐actin/YAP signaling pathway, Journal of Cellular Physiology, 10.1002/jcp.31126, 238, 12, (2751-2764), (2023).
Radstake, W.E., Gautam, K., Miranda, S. et al. Gravitational effects on fibroblasts’ function in relation to wound healing. npj Microgravity 9, 48 (2023)
Minor issues:
2.1. Cell culture and SMG induction Suggest state fluid density and viscosity so readers can calculate the forces if desired.
Author's response:
Thank you for your comment. We have checked the “Safety Data Sheet” of the media for cell culture, however, the information of the Relative density and the viscosity of DMEM/Ham’s F-12 and FBS were not presented. In the next study, we will check fluid density and viscosity of cell culture media before evaluating the effects of SMG on animal cells.
2.2. Suggest state the final concentration of propidium iodide rather than volume added or state the total volume, so concentration can be deduced.
Authors’ response:
In this study, the cell nuclear staining procedure was performed according to the manufacturer's instructions. However, the PI concentration is not presented in the kit. Therefore, we have presented in more detail the nuclear staining procedure for cell cycle assessment by flow cytometry in the materials and methods section.
We hope that our corrections could meet your requirements,
Thank you so much.
Round 2
Reviewer 2 Report
Comments and Suggestions for Authors
The author did not reply the questions clearly and suitablly. The most important point is the morphology of the cells the author used did not like the fibroblasts. The author need to provide the evidence for the cell identificantion.
Comments on the Quality of English LanguageFine.
Author Response
Thank you so much for your comment.
Based on the cell confluency during culture, the 3T3 cell line displays a variety of morphologies. The single cells have a spindle-like appearance. However, they will take on an overlapping swirling shape in the subsequent growth, especially at high density. In this study, the cell density of the SMG group and the control group were 14.04 × 10^3 cells/well and 14.97 × 10^3 cells/well (for 96-well plate), respectively. These results showed that 3T3 cells from both groups get the high confluence after 72 h. This explains the appearance of a swirling shape in 3T3 cells, but not a spindle-like appearance.
We hope that our explanation could meet your requirements,
Thank you so much.
Reviewer 3 Report
Comments and Suggestions for Authors
- The flow cytometry section is sparsely and incomprehensibly written. What method was used to remove cells from the culture tubes? Why were cells fixed with 4% paraformaldehyde for cell cycle analysis? Usually alcohol fixation is used. What protocol was used for cell cycle analysis? How many cells were analyzed by flow cytometry?
Author's response:
After 72 h, the culture medium was discarded and cells were washed twice by 4 ml PBS. Cell detachment was performed with 1 ml trypsin-EDTA 0.25% (TRY-2B, Capricorn Scientific, Ebsdorfergrund, Germany) for 3 min. 3 ml cell culture medium was added to flask and mixed well. Cell suspension was transferred to 15 ml tube and centrifuged at 1500 rpm for 5 min. The medium was discarded and cell pellet was resuspended with 4 ml PBS. Cell suspension was applied for centrifuging at 1500 rpm for 5 min. The medium was discarded and the pellet was resuspended with 4% paraformaldehyde solution (09154-85, Nacalai Tesque, Kyoto, Japan) for 30 min to fix cells. Cell pellet was washed twice by 4 ml PBS and Cell nuclear were then stained with 5 µL propidium iodide (51-66211E, BD Biosciences, San Jose, CA, United States). The cell cycle progression was subsequently analyzed using a flow cytometer BD Accuri C6 Plus (BD Biosciences, San Jose, CA, United States). However,
This response seems unfinished. The Reviewer still considers that described protocol is not suitable for cell cycle analysis. To follow cell cycle progression, it is necessary to stain DNA in all cells. High quality aldehyde fixation guarantees the impermeability of cell membranes.
Author Response
We sincerely apologize for the missing information in the response process. We would like to correct the missing information as below:
After 72 h, the culture medium was discarded and cells were washed twice by 4 ml PBS. Cell detachment was performed with 1 ml trypsin-EDTA 0.25% (TRY-2B, Capricorn Scientific, Ebsdorfergrund, Germany) for 3 min. 3 ml cell culture medium was added to flask and mixed well. Cell suspension was transferred to 15 ml tube and centrifuged at 1500 rpm for 5 min. The medium was discarded and cell pellet was resuspended with 4 ml PBS. Cell suspension was applied for centrifuging at 1500 rpm for 5 min. The medium was discarded and the pellet was resuspended with 4% paraformaldehyde solution (09154-85, Nacalai Tesque, Kyoto, Japan) for 30 min to fix cells. Cell pellet was washed twice by 4 ml PBS and Cell nuclear were then stained with 5 µL propidium iodide (51-66211E, BD Biosciences, San Jose, CA, United States). The cell cycle progression was subsequently analyzed using a flow cytometer BD Accuri C6 Plus (BD Biosciences, San Jose, CA, United States).
However, the results of cell cycle progression by flow cytometry analysis were removed. We just keep cell cycle analysis by Cell Cycle App. of Cytell.
We hope that our corrections could meet your requirements,
Thank you so much.